# Mapping carbon monoxide pollution from space down to city scales with daily global coverage

Tobias Borsdorff[1], Joost aan de Brugh[1], Haili Hu[1], Otto Hasekamp[1], Ralf Sussmann[2], Markus Rettinger[2], Frank Hase[3], Jochen Gross[3], Matthias Schneider[3], Omaira Garcia[4], Wolfgang Stremme[5], Michel Grutter[5], Dietrich G. Feist[6], Sabrina G. Arnold[6], Martine De Mazière[7], Mahesh Kumar Sha[7], David F. Pollard[8], Matthäus Kiel[9], Coleen Roehl[9], Paul O. Wennberg[9,10], Geoffrey C. Toon[11], and Jochen Landgraf[1]

[1]SRON Netherlands Institute for Space Research, Utrecht, the Netherlands
[2]Karlsruhe Institute of Technology (KIT), IMK-IFU, Garmisch-Partenkirchen, Germany
[3]Karlsruhe Institute of Technology (KIT), IMK-ASF, Karlsruhe, Germany
[4]Izaña Atmospheric Research Centre (IARC), Agencia Estatal de Meteorología (AEMET), Santa Cruz de Tenerife, Spain
[5]Centro de Ciencias de la Atmósfera, Universidad Nacional Autónoma de México, México City, México
[6]Max Planck Institute for Biogeochemistry, Jena, Germany
[7]Royal Belgian Institute for Space Aeronomy (BIRA-IASB), Brussels, Belgium
[8]National Institute of Water and Atmospheric Research Ltd (NIWA), Lauder, New Zealand
[9]Division of Geological and Planetary Sciences, California Institute of Technology, Pasadena, CA, USA
[10]Division of Engineering and Applied Science, California Institute of Technology, Pasadena, CA, USA
[11]Jet Propulsion Laboratory (JPL), California Institute of Technology, Pasadena, CA, USA

**Correspondence:** T. Borsdorff (t.borsdorff@sron.nl)

**Abstract.** On 13th October, 2017, the European Space Agency (ESA) successfully launched the Sentinel-5 Precursor satellite with the Tropospheric Monitoring Instrument (TROPOMI) as its single payload. TROPOMI is the first of ESA's atmospheric composition Sentinel missions, which will provide complete long-term records of atmospheric trace gases for the coming 30 years as a contribution to the European Union's Earth Observing programme Copernicus. One of TROPOMI's primary products is atmospheric carbon monoxide (CO). It is observed with daily global coverage and a high spatial resolution of 7x7 km$^2$. The moderate atmospheric resistance time and the low background concentration leads to localized pollution hot spots of CO and allows to track the atmospheric transport of pollution on regional to global scales. In this contribution, we demonstrate the groundbreaking performance of the TROPOMI CO product, sensing CO enhancements above cities and industrial areas and tracking, with daily coverage, the atmospheric transport of pollution from biomass burning regions. The CO data product is validated with two months of Fourier-transform spectroscopy (FTS) measurements at nine ground-based stations operated by the Total Carbon Column Observing Network (TCCON). We found a good agreement between both data sets with a mean bias of 6 ppb (average of individual station biases) for both clear-sky and cloudy TROPOMI CO retrievals. Together with the corresponding standard deviation of the individual station biases of 3.8 ppb for clear-sky and 4.0 ppb for cloudy-sky, it indicates that the CO data product is already well within the mission requirement.

# 1 Introduction

The Sentinel-5 Precursor (S5P) satellite was successfully launched on 13th October, 2017, from Plesetsk in northern Russia with the Tropospheric Monitoring Instrument (TROPOMI) aboard. The instrument is a grating spectrometer which measures sunlight reflected by the Earth's atmosphere and its surface from the ultraviolet to the shortwave infrared (SWIR) with daily global coverage, a spatial resolution of about 7x7 km$^2$, and a high radiometric accuracy to infer the CO total column over dark vegetation surfaces with an precision $< 10$ % (Veefkind et al., 2012). One of the primary targets of the mission is to monitor the atmospheric concentration of CO. This trace gas is emitted to the atmosphere by incomplete combustion e.g. by traffic, industrial production, and biomass burning. Its major sink is the reaction with the OH radical (Spivakovsky et al., 2000). With a typical background concentration of ca. 80 ppb (in the Northern Hemisphere) and an atmospheric residence time from days to months (Holloway et al., 2000), the trace gas is established as a tracer of how pollution is transported, redistributed, and depleted in the atmosphere.

The S5P mission builds upon the heritage of SCIAMACHY (Scanning Imaging Absorption Spectrometer for Atmospheric Chartography; Bovensmann et al. (1999)), which provided atmospheric CO total column concentrations from the same spectral range (Gloudemans et al., 2009; Frankenberg et al., 2005; Buchwitz et al., 2007; Gimeno Garcia et al., 2011; Borsdorff et al., 2017a)). Measurements of SCIAMACHY in the SWIR have a spatial resolution of about 30 km $\times$ 120 km (along-track $\times$ across-track) for an integration time of 0.5 s with a global coverage cycle of 3 days. Most importantly, the SCIAMACHY noise error of single CO retrievals can exceed 100 % for dark scenes. Hence, spatial and temporal averaging of single CO measurements is required (de Laat et al., 2007; Gloudemans et al., 2006), which limits the data interpretation of SCIAMACHY CO data.

From space, CO is also measured by other satellite instruments with global coverage, e.g. MOPITT (Measurements of Pollution in the Troposphere; Deeter (2003)), AIRS (Atmospheric Infrared Sounder; McMillan (2005)), TES (Tropospheric Emission Spectrometer; Rinsland et al. (2006)), IASI (Infrared Atmospheric Sounding Interferometer; Turquety et al. (2004)). The S5-P mission is the first of a sequence of ESA's atmospheric composition satellites, which also comprises the Sentinel-5 mission, a series of spectrometers with the first launch in the 2021-2023 time frame.

In this study, we use the Shortwave Infrared CO Retrieval Algorithm (SICOR). It is developed by SRON the Netherlands Institute for Space Research, for the operational processing of TROPOMI data (Landgraf et al., 2016a, b) and also serves as algorithm baseline for the data processing of the successor mission Sentinel 5. The algorithm infers the vertical column concentration of CO (the vertically integrated amount of CO above the surface) simultaneously with effective cloud parameters from TROPOMI's 2.3 $\mu$m spectra (Vidot et al., 2012; Landgraf et al., 2016b). A first comparison of the TROPOMI CO data product with CO fields from the European Center for Medium-Range Weather Forecast (ECMWF) was performed by Borsdorff et al. (2018). Based on this, our study deploys SICOR on TROPOMI measurements taken during the early months of the instrument in orbit, and show the capability of the instrument to detect and monitor the air pollution from hot spots like larger cities and industrial regions. Investigating the temporal evolution of CO enhancements over the Atlantic and Indian oceans shows the capability of the instrument to track the atmospheric transport of pollution on a day-to-day basis in agreement with

co-located ground-based measurements. Moreover, a validation with collocated ground-based Fourier Transform Spectrometer (FTS) measurements at nine TCCON sites, indicates the TROPOMI CO data quality. The paper is structured as follows: Section 2 describes the data set and methodology and Sec. 3 presents our analysis of the TROPOMI CO data product comprising a first validation with TCCON ground-based measurements, the detection of CO hot spots and the transport of CO pollution over the oceans. Section 4 gives the conclusions of the study and finally the data availability is described in Sec. 5.

## 2   Data set and methodology

For this study, we used nominal Earth radiance and Solar irradiance measurements of TROPOMI during the commissioning phase of the instrument from 9th November, 2017 to 4th January, 2018. We deployed the SICOR algorithm on the 2.3 $\mu$m spectra of TROPOMI and retrieved the total column density of CO simultaneously with interfering trace gases and effective cloud parameters (cloud height ($z$) and optical thickness ($\tau$)) describing the cloud contamination of the ground scene (Landgraf et al., 2016b). The retrieval approach is based on the profile scaling method (Borsdorff et al., 2014) and the implementation and retrieval settings are discussed in detail by Landgraf et al. (2016a). The reference profile of CO that is scaled during the retrieval is taken from simulations of the global chemical transport model TM5 (Krol et al., 2005) and monthly averaged over 3 degree $\times$ 2 degree latitude/longitude grid boxes. Hence, the retrieval result is the total column density of CO [molec/cm$^2$]. To compare it with other measurements we also represent the data product as a dry column mixing ratio XCO [ppb] by dividing the CO total column density by the dry air column density derived from colocated ECMWF pressure fields.

For the data analysis, we performed *a posteriori* quality filtering of the TROPOMI data. To this end, we used retrievals with a solar zenith angle < 80 degrees and discarded the two most westward ground pixels of the swath due to a performance issue that is still under investigation. Furthermore, we distinguished between retrievals under clear-sky ($\tau < 0.5$ and $z < 5$ km, over land) and cloudy condition ($\tau > 0.5$ and $z < 5$ km, over land and ocean). The remaining retrievals are not considered in this study. Clear-sky observations over ocean, which is a dark surface in the SWIR, cannot be used for data interpretation because of a too low signal.

For the total column of CO, the vertical sensitivity of the retrieval is described by the total column averaging kernel (Borsdorff et al., 2014), which is illustrated in Fig. 1 for TROPOMI data of one particular day, 10th November 2017. It shows the dependence of the averaging kernel on the cloudiness of the scene, where the standard deviation indicates its variation due to different observation and atmospheric parameters, e.g. solar zenith angle, viewing zenith angle and ground reflectivity. For very strict cloud clearing of the data (with $z < 5$ km and $\tau < 0.01$), the total column averaging kernel is close to 1 for all altitudes with little variation, meaning that the derived column can be interpreted as an estimate of vertically integrated amount of CO. Filtering the data less strict using the clear-sky filter from above ($z < 5$ km and $\tau < 0.5$) results in a slightly reduced sensitivity with a moderate standard deviation and Borsdorff et al. (2017a) concluded that those measurements are usually clear-sky equivalent for remote regions without local pollution sources and the induced errors due to the choice of the reference profile to be scaled by the inversion to be on a percentage level (Borsdorff et al., 2014). The presence of clouds changes significantly the vertical sensitivity of the retrieval. Figure 1 shows the column averaging kernel when filtering for optical thick clouds at 5 km

altitude. The sensitivity below the cloud is significantly reduced (values lower than 1) due to cloud shielding, and the retrieval estimates a CO total column mainly based on the measurement sensitivity to CO above the cloud (values higher than 1).

Consequently, the direct comparison of reference measurements with the retrieved CO columns from cloud contaminated TROPOMI measurements can lead to errors > 30% (Borsdorff et al., 2014). This so called smoothing error is due to imperfect knowledge of the vertical profile of CO. However, the TROPOMI CO dataset provides total column averaging kernels $a_col$ for each retrieval. To compare a vertical profile $\rho$ e.g. from airborne in-situ measurements or model simulations with the TROPOMI CO product a total column concentration $c = a_{col}\dot{\rho}$ can be calculated from $\rho$. This can be directly compared with the retrieval result since it is in the same way affected by the reduced sensitivity as the retrieval (Rodgers and Connor, 2003). When the reference measurement is not a vertical profile the application of the total column averaging kernel becomes more difficult. In that case, the TROPOMI CO dataset can be filtered for retrievals under clear-sky conditions to avoid misinterpretations. Alternatively, an approach as presented by Cogan et al. (2012) can be followed who quantified expected differences in GOSAT/TCCON $CO_2$ retrievals due to averaging kernel differences using the GEOS-Chem model to simulate a realistic range of $CO_2$ profiles.

The TROPOMI instrument is still in the early phase of the mission and the performance of the CO retrieval is expected to improve in the future. For example, single overpasses show stripes of erroneous CO in flight direction, probably due to calibration issues of TROPOMI. Considering high-frequency variations of CO measurements across flight direction per orbit, we infer the stripe pattern by median filtering of the detected features in flight direction per orbit. Figure 2 provides an example, where the average the average of the stripe pattern in cross flight direction is -0.03 ppb with a standard deviation of 1.1 ppb. Some stripes can reach values higher than 5 ppb. Hence, the stripe pattern can be removed from the data a posteriori to the retrieval and its removal is indicated accordingly in the remainder of the paper. Boersma et al. (2011) suggested a similar approach to improve the quality of the $NO_2$ data product of the Ozone Monitoring Instrument (OMI, Levelt et al. (2006)).

## 3 Results

### 3.1 Validation with TCCON ground-based measurements

The quality of the TROPOMI CO data product needs to be validated with independent reference observations both for clear-sky and cloudy TROPOMI measurements. To this end, we performed a first validation with CO observations at nine ground-based FTS stations operated by the TCCON network (see Table 1) which are located preferably at remote areas.

TCCON is a network of ground-based Fourier Transform Spectrometers to measure total column concentrations of atmospheric trace gases including CO with high accuracy and precision e.g. for satellite validation. The trace gas columns are retrieved from spectrally highly resolved near-infrared radiance measurements recorded in direct-sun geometry (Wunch et al., 2015). Cloud contaminated measurements are rejected and so TCCON measurements refer to clear-sky observations only. Here, TCCON CO columns are provided as column averaged dry air mole fractions XCO (Wunch et al., 2010).

We selected sites in both the northern and southern hemisphere at low and high elevation on the continents and islands (Hase et al., 2015; Sussmann and Rettinger, 2014; Wennberg et al., 2016, 2015; Blumenstock et al., 2014; Feist et al., 2014; De

Mazière et al., 2014; Sherlock et al., 2014). Wunch et al. (2015) reported that the total error of the XCO columns measured by TCCON is below 4%. This also includes an estimation of the smoothing error that is about 1% for the TCCON CO product. The magnitude of the smoothing error was assessed by changing the shape of the reference profile used for the TCCON scaling retrieval. Within this error margin we can assume the TCCON measurements as an estimate of the truth.

For the comparison, we used TROPOMI observations co-located with the TCCON sites by selecting all TROPOMI retrievals from the same day within a radius of 50 km around each station. The retrieved CO column of TROPOMI is adapted to the altitude of the station by either cutting off the scaled mixing ratio profile at the station altitude or extending it assuming a constant elongation of the mixing ratio to lower altitude. For mountain stations like Zugspitze and Izana, this reduces the TROPOMI CO column on average by 10 and 4 ppb, respectively, improving the agreement between the ground-based and

satellite measurements accordingly. Finally, we calculated daily averages of the XCO values using the adapted TROPOMI retrievals and the TCCON measurements shown in Fig. 3 and Fig. 4. Data gaps in the TROPOMI time series are partly caused by discarding observations with high clouds (>5 km) but also due to observation time reserved for in-orbit instrument characterization during the instrument commissioning phase.

Figure 5 depicts the corresponding bias for each TCCON station for clear-sky and cloudy-sky conditions and the combination

of both, as well as the standard deviation and the number of coincident daily mean values of TROPOMI and TCCON. With the limited data available at the time of writing, we found good agreement with a small mean bias of TROPOMI CO versus TCCON of 6.0 ppb for clear-sky, 6.2 ppb for cloudy-sky TROPOMI retrievals and 5.8 ppb for the combination of both with a standard deviation of the individual station biases of 3.8 ppb for clear-sky, 4.0 ppb for cloudy-sky, and 3.4 ppb for the combination case. Furthermore, the mean standard deviation of the bias is 3.9 ppb for clear-sky, 2.4 ppb for cloud-sky, and 2.9

ppb for the combination. The good agreement between clear-sky and cloudy-sky retrieval underlines the validity of the data retrieval for cloudy scenes, a key aspect of the SICOR algorithm to achieve the data coverage of the TROPOMI CO product.

Most of the TCCON stations are only affected by remote pollution sources, this explains the good agreement between the validation of the clear-sky and cloudy-sky TROPOMI retrievals. This may differ in the presence of local pollution sources where the shape of the under-cloud CO profile can strongly deviate from the one of the reference profile used for the profile

scaling of the TROPOMI CO dataset. In such cases, when the TROPOMI CO data set is directly compared with reference measurements without applying the averaging kernel clear-sky are always preferable compared to cloudy-sky observations.

## 3.2   Detection of CO hot spots

Today's work and life style supports urbanization and the rise of metropolitan areas all over the world with populations exceeding more than 10 million people. Intense traffic and industrial activities in those regions lead to high levels of air pollution

affecting human health. For example, at rush hour in Mexico City the CO concentration has reached values as high as 9300 ppb (Stephens et al., 2008). Sensing air pollution from space has the potential to globally monitor trends and variations of atmospheric pollutants affecting human health. The detection of air pollution above cities, urban and industrial areas with satellites comes with the challenge of low CO sensitivity of measurements. Until now, data needs to be temporally and spatially aver-

aged to distinguish typical the CO enhancements of $\leq 20$ ppb of the total column dry air mixing ratio from the surrounding background concentrations in the order of 100 ppb (Pommier et al., 2013; Clerbaux et al., 2008; Borsdorff et al., 2017b).

In this respect, the CO measurements by TROPOMI represent a breakthrough. The advanced radiometric performance combined with the high spatial resolution and the daily global coverage of TROPOMI allows the sensing of CO enhancements above polluted areas with only single satellite overpasses, given the perspective of day-to-day monitoring. For example, Fig. 6a shows enhanced CO values over the industrial area near to Venice as well as pollution above Turin, Milan, and Rome. Figure 6b depicts an orbit overpass over Saudi Arabia and Egypt and shows distinct pollution patterns over Mecca, Jeddah and Cairo. Furthermore, enhanced CO values along the Nile indicate air pollution in this densely populated region. Figure 6c clearly shows the enhanced CO values above Tehran, in agreement with the urban area of the city. Also smaller cities in the region like Qom, Isfahan, and Mashhad can be distinguished from the background CO level. Finally, Fig. 6d shows strong CO enhancements above Mexico City, Guadalajara, Torreón, and Monterrey. Data gaps in the figures are caused by the filtering of measurements under clear-sky conditions over the oceans and measurements contaminated by high altitude clouds. Figure 6 shows predominately clear-sky observations but also includes retrievals from cloud contaminated scenes, which in case of optical thick and high clouds reduces the sensitivity to boundary layer CO pollution at emission hot spots (Borsdorff et al., 2017b). Neither temporal nor spatial averaging is necessary to distinguish the CO enhancements of the total column above the shown point sources. The average noise error of the retrievals from the individual ground pixels shown in Fig. 6 is $< 2.3$ ppb.

The daily global coverage of TROPOMI and so the temporal evolution of air pollution on city scales opens up new possibilities to monitor the effect of emission regulations but also requires estimates of the absolute uncertainty of the TROPOMI CO product. Figure 7a shows that the TROPOMI CO concentrations are in good agreement with ground-based measurements of a Fourier Transform Spectrometer (FTS) in Mexico City (Stremme et al., 2009, 2013; Plaza-Medina et al., 2017) when selecting clear-sky satellite observations of the same day, which are spatially co-located in a radius of 15 km around the ground site. Figure 7b indicates that this data screening is essential for the detection of pollution on city scales. Choosing a wider co-location radius for the satellite data leads to a significant bias with the FTS measurements, which demonstrates the importance of TROPOMI's spatial resolution for this type of application keeping representation errors between ground-based and satellite observations to a minimum. It is important to realize that the comparison with measurements at TCCON sites, as discussed in the previous subsection, are mostly not affected by localized CO emissions and so allow for a looser spatial collocation criterion with a collocation radius of 50 km.

Furthermore, selecting TROPOMI CO observations for cloudy conditions leads to a 25 % (5e17 molec/cm$^2$) bias independent from the selected radius. Due to light shielding by clouds, the satellite measurements become insensitive to the lower atmosphere where most of the pollution is located (Borsdorff et al., 2014) and so the TROPOMI CO product estimates the CO column from the less polluted air above the cloud. This apparent disadvantage of cloudy observations turns into an advantage when analyzing the vertical distribution of trace gases (Borsdorff et al., 2017b). By observing the same pollution event for clear-sky conditions and for varying cloud height, it reveals the vertical extension of the city pollution into the atmosphere.

### 3.3 Monitoring pollution transport

For several years, measurements of CO have been used to trace the transport of polluted air masses within the atmosphere, mostly with the focus on long-range transport. The atmospheric residence time of CO varies from days to months (Holloway et al., 2000) and so it is well suited to capture advection of atmospheric pollution. For example, Gloudemans et al. (2006) studied the transport of CO emission by biomass burning in the southern hemisphere using SCIAMACHY observations and Yurganov et al. (2004, 2005) analyzed the anomaly in the CO burden of the northern hemisphere caused by biomass burning with ground-based and satellite measurements.

The TROPOMI CO data set will advance this research field by providing the global distribution of the atmospheric CO concentration on a daily basis with high spatial resolution. It enables us to study the day-to-day variation of CO on global, regional and local scales. As an example, Fig. 8a-c present TROPOMI CO for three subsequent days depicting the southward transport of enhanced CO concentrations over the Atlantic Ocean originating from fires in north Africa. On 13th December, 2017 Ascension Island is surrounded by air with low CO concentrations (80 ppb) but already a few days later, on 17th December, the first enhanced CO values reach the island. Finally on December 25th, Ascension Island is exposed to strong CO polluted air with values up to 116 ppb. Figure 4 (panel 3) shows that this finding is in agreement with regular ground-based FTS measurements of a Total Carbon Column Observing Network (TCCON) station on the island (Feist et al., 2014). With the help of TROPOMI measurements the localized ground-site measurements can be put into a regional context. Another example is given in Fig. 8d-f. On November 10th, biomass burning in Africa and Madagascar caused an extended plume of enhanced CO concentrations reaching the island of Réunion. The atmospheric situation stayed stable until November 12th (90-103 ppb), but changed to low CO concentration on the 16th of November due to different meteorology (70 ppb). Figure 4 (panel 4) shows an excellent agreement with TCCON measurements at the island (De Mazière et al., 2014) and so illustrates the extra information provided by the satellite product in addition to the ground-based measurement.

### 4 Conclusions

The CO observations of TROPOMI represent a major step forward for the monitoring of air pollution from space. In this study we investigated the quality of the CO data product applying the operational retrieval software to the first two months of data of the S5-P mission in space. Because only a small fraction of all TROPOMI observations are expected to be cloud free (Krijger et al., 2005), it is essential to account for clouds in the retrieval. Building on previous work (Gloudemans et al., 2009; Buchwitz et al., 2006; Vidot et al., 2012), an important feature of the TROPOMI CO retrieval is to infer cloud parameters and trace gas columns at the same time to achieve a good yield from the data processing. Thereby, we account for the vertical sensitivity of cloudy measurements and enhance the data yield both for observations over land and ocean (Borsdorff et al., 2017a, 2016). On average 8% of all measurements are clear-sky, 22 % cloudy-sky observations over land, and 51% cloud-sky over the oceans.

A first validation of the TROPOMI CO data product with collocated TCCON observations at nine selected measurement sites showed good agreement with a mean bias of about 6 ppb for both clear-sky and cloudy observations and a mean standard deviation of 3.9 ppb and 2.4 ppb, respectively, demonstrating a good repeatability of the observations. Here, the standard

deviation of the station biases is 3.8 ppb and 4.0 ppb for both types of measurements. Additionally, a comparison with ground-based FTS at Mexico shows that the CO pollution hot spot can be observed with high accuracy. For this study, only a limited amount of TCCON data was available with confined spatial and temporal coverage. The Sentinel 5 Precursor as an operational mission requires a continuous monitoring of the CO data quality, which will be performed as part of the operational validation

activities. In this context, future work will consider the validation of the TROPOMI CO data for longer time scales including additional TCCON and NDACC stations to improve the significance of the product validation.

Due to the good data quality in combination with the spatial resolution of 7x7 km$^2$, TROPOMI can capture the variability of CO due to atmospheric transport of pollution on a day-to-day basis, demonstrated for daily overpasses at Ascension Island and Reunion in agreement with TCCON observations at these sites. Based on these preliminary results, we conclude that the

TROPOMI CO product already fulfills the mission requirements on precision ($< 10\,\%$) and accuracy ($< 15\,\%$) (Veefkind et al., 2012), assuming a background concentration of 100 ppb.

## 5    Data availability

The TROPOMI CO data set of this study is available for download at ftp://ftp.sron.nl/pub/pub/DataProducts/TROPOMI_CO/. The underlying data of the figures presented in this publication can be found at ftp://ftp.sron.nl/open-access-data/. TCCON

data are available from the TCCON Data Archive, hosted by CaltechDATA, California Institute of Technology, CA (US), https://tccondata.org/. The TROPOMI CO data is available via the Copernicus Open Access Hub https://s5phub.copernicus.eu.

*Author contributions.*  Tobias Borsdorff, Joost aan de Brugh, Haili Hu, and Otto Hasekamp have done the TROPOMI CO retrieval and data analysis. Ralf Sussmann, Markus Rettinger, Frank Hase, Jochen Gross, Matthias Schneider, Omaira Garcia, Wolfgang Stremme, Michel Grutter, Dietrich G. Feist, Sabrina G. Arnold, Martine De Mazière, Mahesh Kumar Sha, David F. Pollard, Matthäus Kiel, Coleen Roehl, Paul

O. Wennberg, and Geoffrey C. Toon performed FTS measurements and retrievals for the various stations. Jochen Landgraf supervised the study. All authors discussed the results and commented on the manuscript.

*Competing interests.*  The authors declare no competing interests.

*Disclaimer.*  The presented work has been performed in the frame of the Sentinel-5 Precursor Validation Team (S5PVT) or Level 1/Level 2 Product Working Group activities. Results are based on preliminary (not fully calibrated/validated) Sentinel-5 Precursor data that will still

change.

*Acknowledgements.* We would like to thank the team that has realized the TROPOMI instrument, consisting of the partnership between Airbus Defense and Space, KNMI, SRON and TNO, and commissioned by the Netherlands Space Office (NSO) and the European Space Agency (ESA). In particular, we acknowledge Ilse Aben and Ruud Hoogeveen, the SRON L1 team. Sentinel-5 Precursor is part of the EU Copernicus programme. Sentinel-5 Precursor is a ESA mission on behalf of the European Commission (EC). The TROPOMI payload is a joint development by ESA and the NSO. The Sentinel-5 Precursor ground-segment development has been funded by ESA and with national contributions from The Netherlands, Germany, and Belgium. The TCCON site at Réunion Island is operated by the Royal Belgian Institute for Space Aeronomy with financial support in 2014, 2015, and 2016,2017,2018 under the EU project ICOS-Inwire and the ministerial decree for ICOS (FR/35/IC2) and local activities supported by LACy/UMR8105 – Université de La Réunion. The Belgian co-authors are also supported by the PRODEX TROVA project. The measurements in Mexico City are founded by the projects CONACYT (No. 275239 and No. 239618) and UNAM-DGAPA-PAPIIT (No. IN112216 and No. IN111418) Alfredo Rodrigez, Miguel Angle Robles, Delibes Flores Roman, Wilfrido Gutiérrez and Alejandro Bezanilla are acknowledged for technical support. This research has been funded in part by the TROPOMI national program from the NSO. The TROPOMI data processing was carried out on the Dutch national e-infrastructure with the support of the SURF Cooperative. The operation of the Ascension Island TCCON site was funded by the Max Planck Society.

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

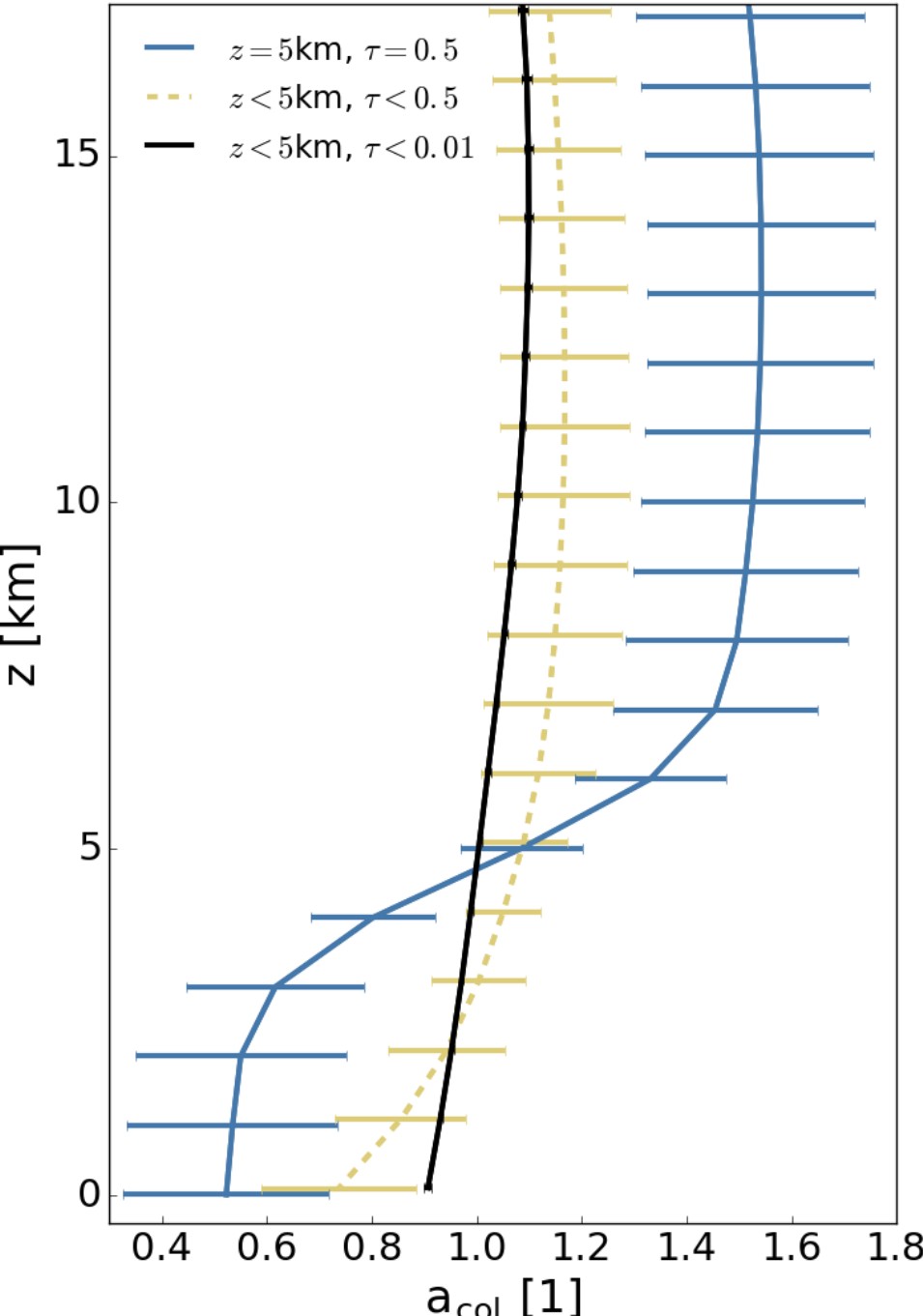

**Figure 1.** TROPOMI CO total column averaging kernels for November the 10th, 2017. The global average is shown for three different categories of cloudiness (black) strict cloud clearing, (yellow) clear-sky equivalent, and (blue) high optical thick clouds. The standard deviation is indicated as error bars.

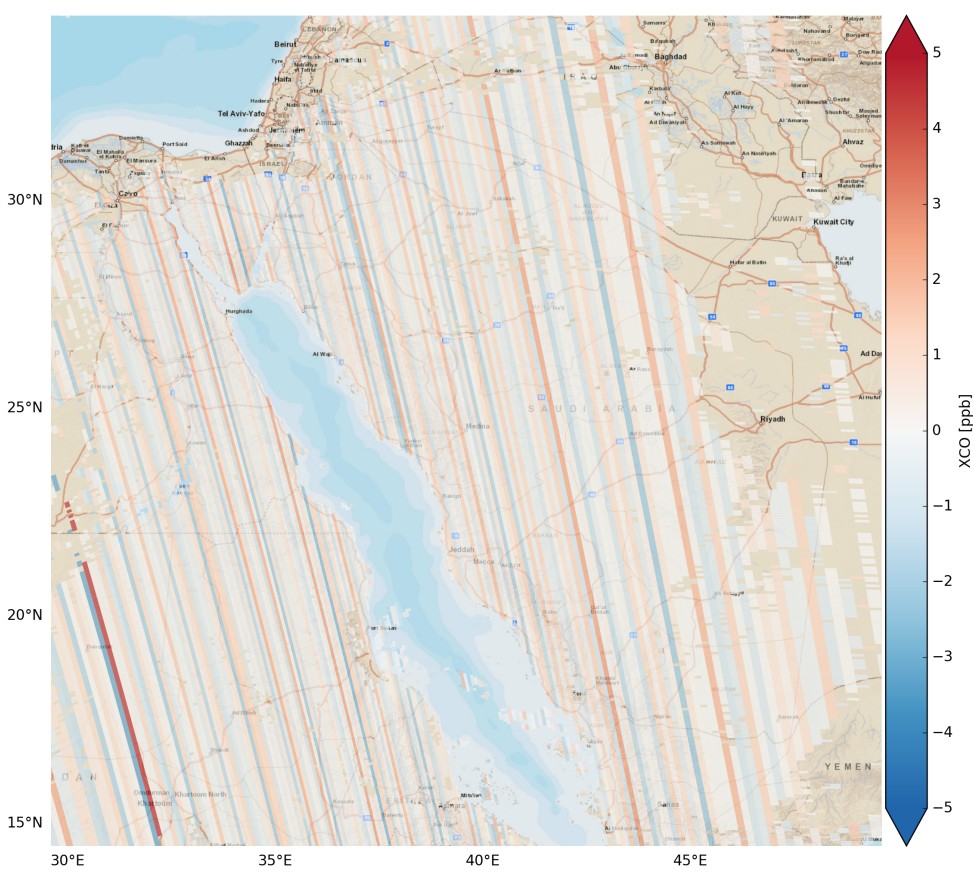

**Figure 2.** Stripe pattern derived by median filtering from a TROPOMI CO orbit above Saudi Arabia and Egypt on 12th November 2017.

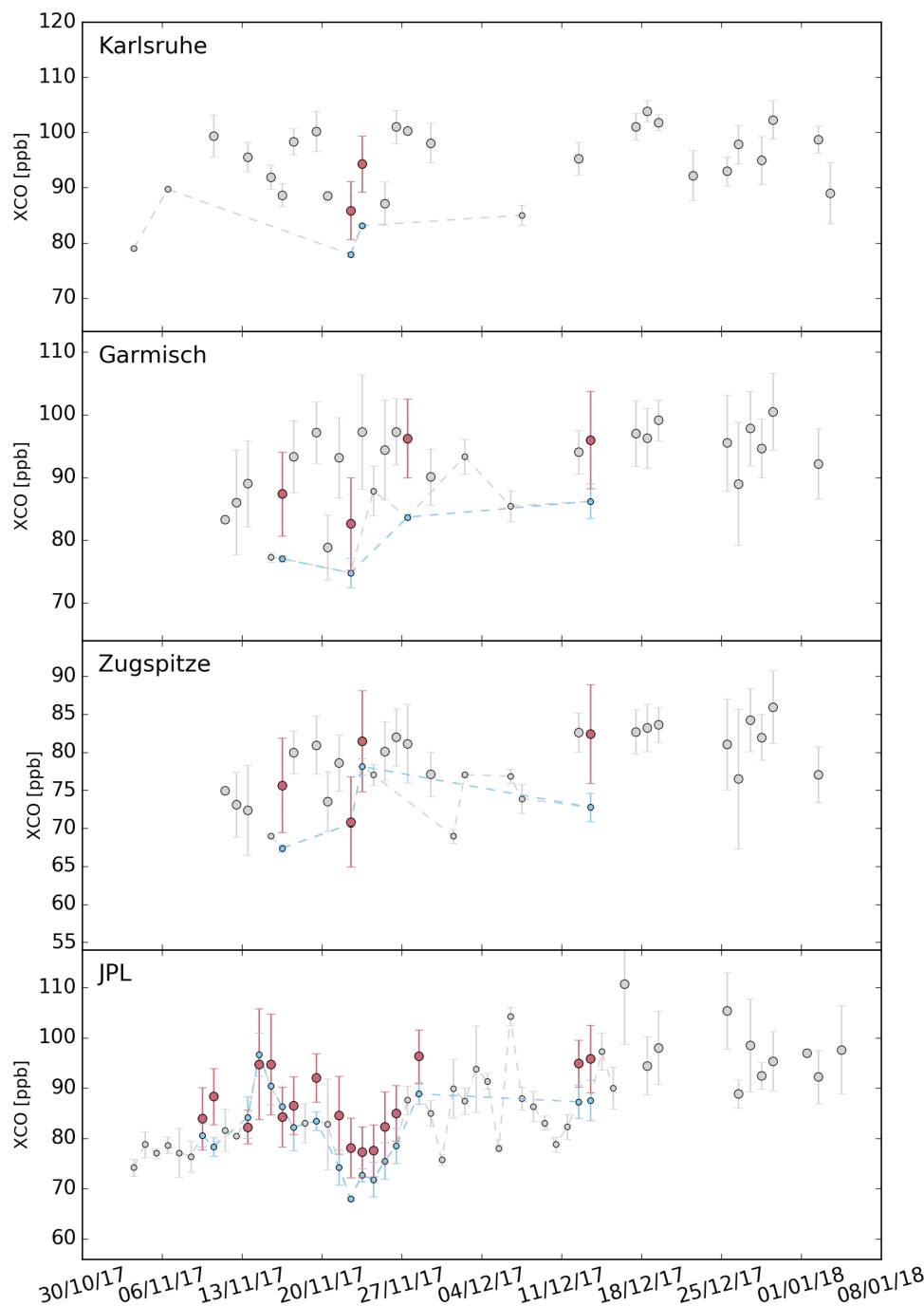

**Figure 3.** Daily means of dry air column mixing ratios (XCO) measured by TROPOMI (pink) and various TCCON stations (blue). A co-location radius of 50 km is used. The standard deviation of individual retrievals within a day is shown as an error bar. Data points without time coincidence between TCCON and TROPOMI are plotted in grey. No de-striping was applied on the TROPOMI data.

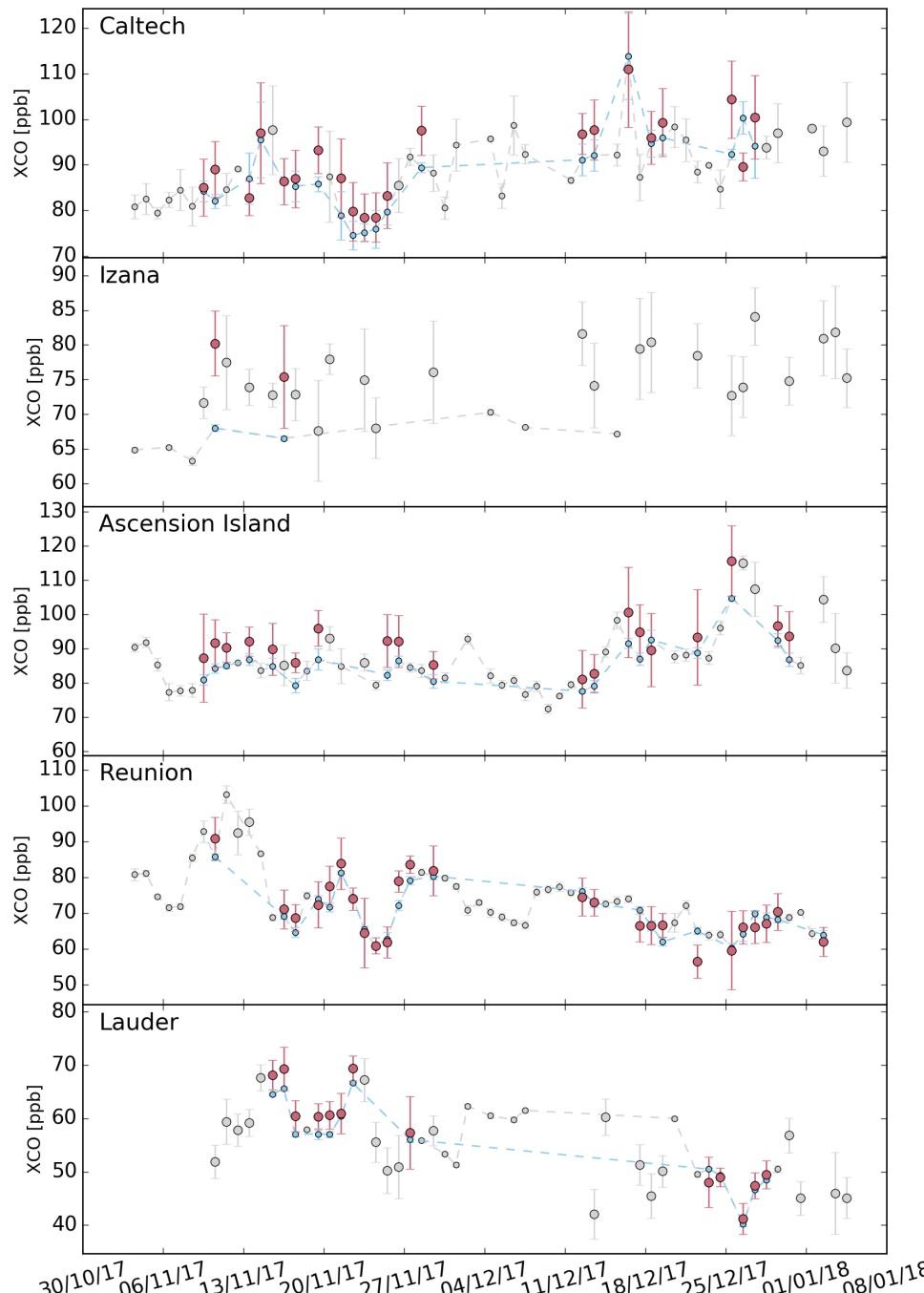

**Figure 4.** As Fig. 3 but with different TCCON stations.

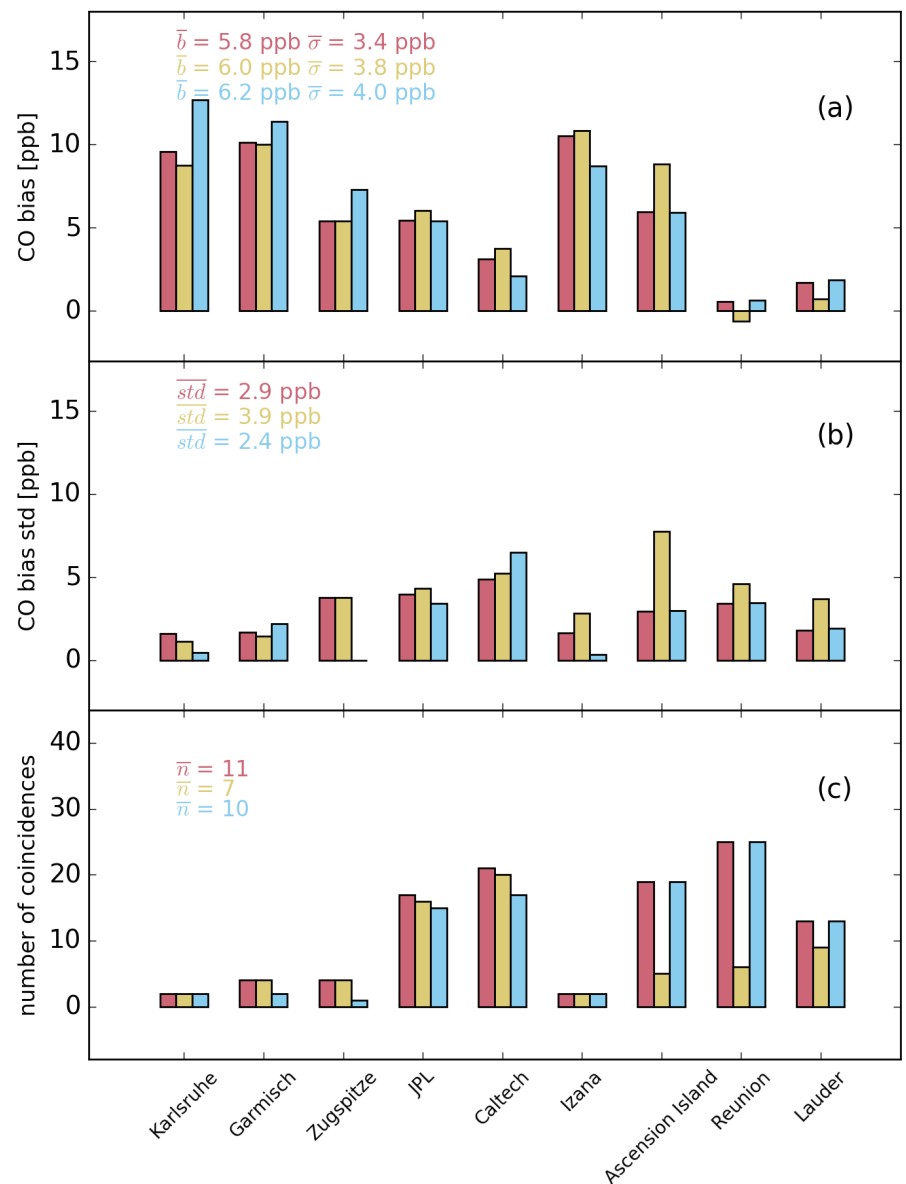

**Figure 5.** Mean bias (TROPOMI - FTS) between co-located daily mean XCO values (see Fig. 3, 4) of TROPOMI and TCCON (a), the standard deviation of the bias (b), and the number of coincident daily mean pairs (c). $\bar{b}$ is the global mean bias (average of all station biases) and $\bar{\sigma}$ its standard deviation . $\overline{std}$ is the average of all standard deviations and $\bar{n}$ the average number of coincident pairs. TROPOMI retrievals under clear-sky (yellow), cloudy-sky (blue) and the combination of both (pink) are distinguished. No de-striping was applied on the TROPOMI data.

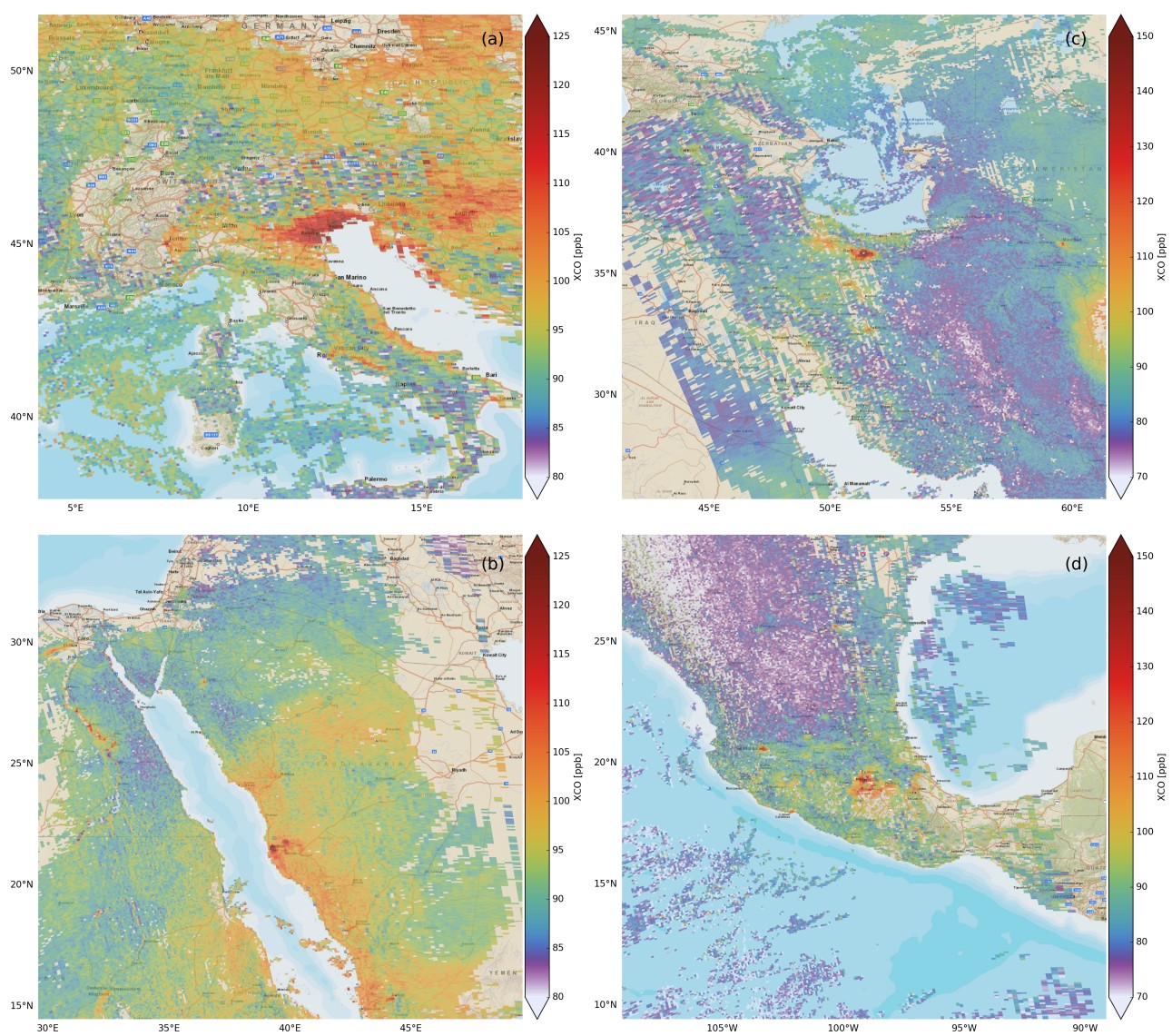

**Figure 6.** Total column mixing ratio (XCO) for individual TROPOMI ground pixels for (a) Italy on 25th December, (b) Saudi Arabia and Egypt on 12th November 2017, (c) Iran on 17th November 2017, and (d) Mexico on 25th November 2017. De-striping was applied on the TROPOMI data.

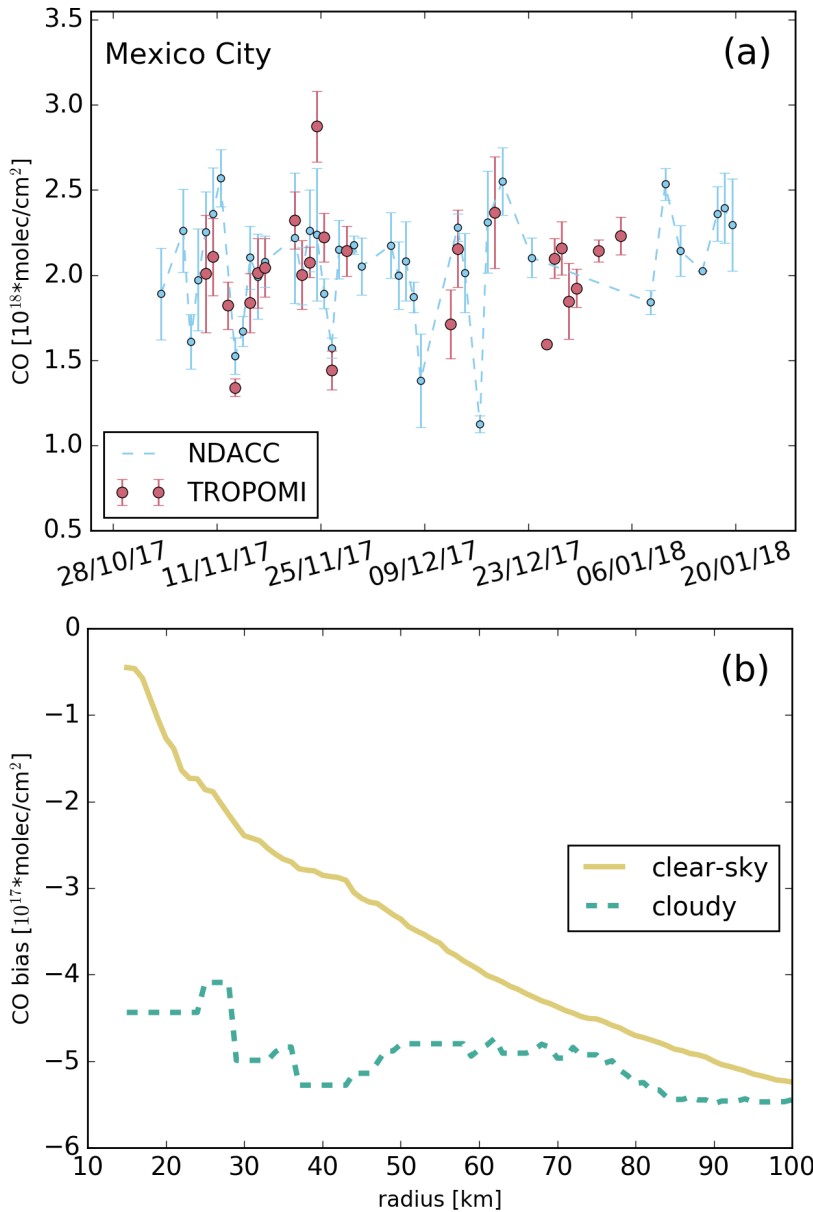

**Figure 7.** (a) Mexico City daily mean CO columns from TROPOMI (pink) and FTS (blue) with standard deviation of the individual retrievals (errors). TROPOMI observations are filtered for clear-sky within 15 km around the ground site. The TROPOMI columns are altitude corrected to the station elevation. (b) Bias of the CO columns (TROPOMI - FTS) as function of the co-location radius around Mexico City. Here, TROPOMI retrievals under clear-sky conditions (yellow) and optically thick clouds above 4000 m (green) are considered. For the smallest radius (15 km) we found 20 cloudy and 160 clear-sky collocations. However, for the widest radius (100 km) 92 cloudy and 4425 clear-sky collocations are found. No de-striping was applied on the TROPOMI data.

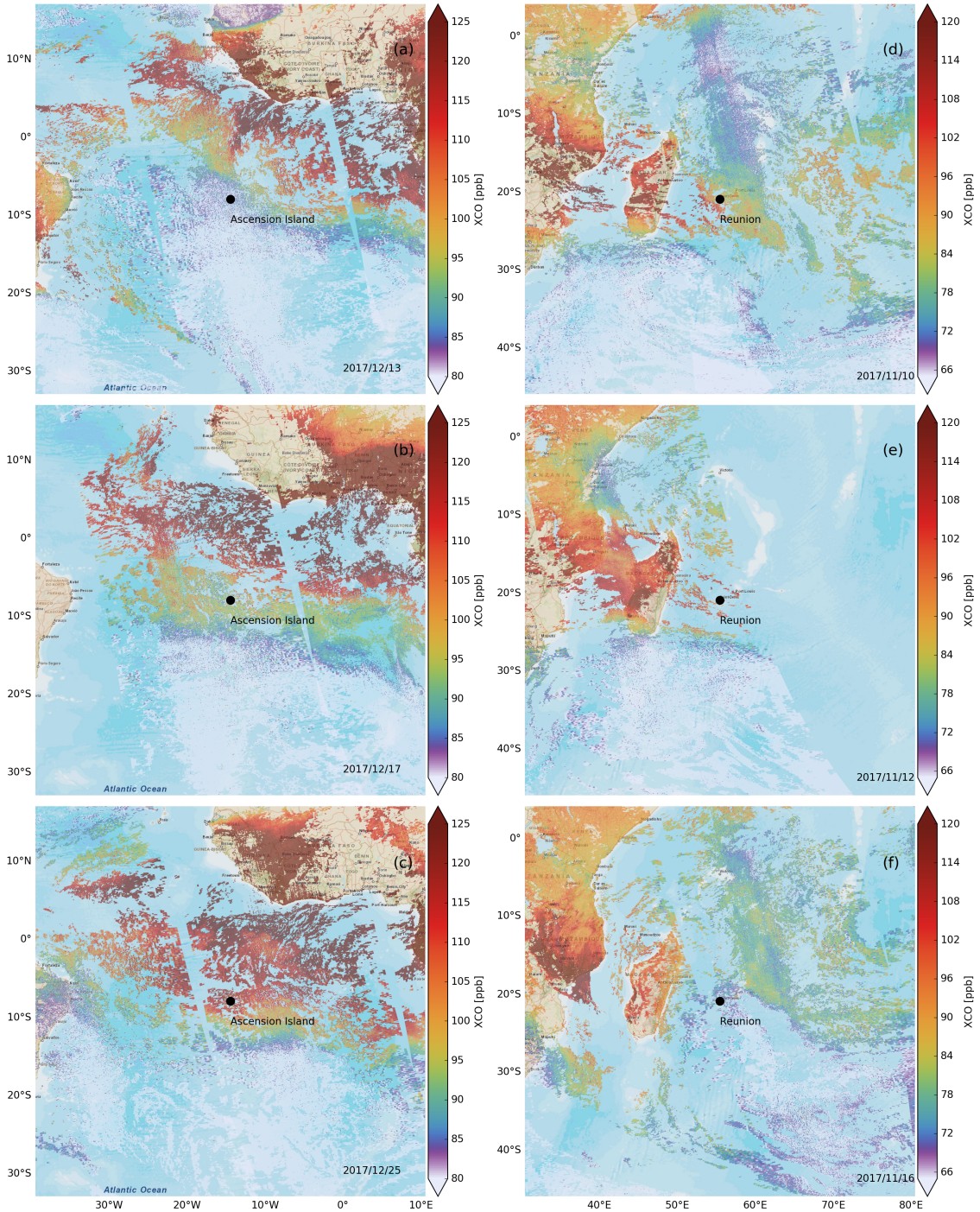

**Figure 8.** Total column mixing ratio (XCO) for individual TROPOMI ground pixels near Ascension Island on (a) 13th December, 2017, (b) 17th December, 2017, and (c) 25th December, 2017 and near Réunion on (d) the 10th November, 2017 (e) 12th November,2017, and (f) 17th November, 2017. De-striping was applied on the TROPOMI data.

**Table 1.** Ground-based FTS stations used for validation. The latitude and longitude values are given in degrees, the surface elevation in km.

| Name | Latitude | Longitude | Altitude | Type |
|---|---|---|---|---|
| Karlsruhe | 49.10 | 8.44 | 0.11 | TCCON |
| Garmisch | 47.48 | 11.06 | 0.75 | TCCON |
| Zugspitze | 47.42 | 10.98 | 2.96 | TCCON |
| JPL | 34.20 | −118.18 | 0.39 | TCCON |
| Caltech | 34.14 | −118.13 | 0.24 | TCCON |
| Izaña | 28.31 | −16.50 | 2.37 | TCCON |
| Mexico City | 19.33 | −99.18 | 2.26 | Bruker Vertex 80 |
| Ascension Island | −7.92 | −14.33 | 0.03 | TCCON |
| Réunion | −20.90 | 55.49 | 0.09 | TCCON |
| Lauder | −45.04 | 169.68 | 0.37 | TCCON |