# Peer review of "Mapping carbon monoxide pollution from space down to city scales with daily global coverage"

_Atmospheric Measurement Techniques, 2018_

## Referee Comment (RC1) · Anonymous Referee #1 · 1 Jun 2018

**General Comments**

This brief manuscript reports early findings from CO retrievals by the TROPOMI instrument launched in late 2017. In that sense, the manuscript is similar to a GRL paper ('Measuring Carbon Monoxide With TROPOMI: First Results and a Comparison With ECMWF-IFS Analysis Data') recently published by Borsdorff et al. The main novel aspect of this new manuscript is an intercomparison with ground-based CO measurements from nine TCCON stations. The manuscript also presents several short 'case studies' demonstrating CO gradients associated with urbanization and long-range transport.

Readers of Atmospheric Measurement Techniques may find this paper interesting, although it contains little technical information. Since the recent GRL paper by Borsdorff et al. already demonstrated that the TROPOMI instrument and SICOR retrieval algorithm yield physically reasonable CO retrievals, this manuscript should really go into the quantitative use of the TROPOMI CO product in some depth. Without such detailed technical information, users will not be able to fully exploit this valuable new dataset. Some of this information is contained in the 2014 AMT paper by Borsdorff et al. ('Insights into Tikhonov regularization ...') and the 2016 AMT paper by Landgraf et al. ('Carbon monoxide total column retrievals from TROPOMI ...'), although the relevance of those theoretical papers to operational TROPOMI retrievals (which are inevitably more complicated to analyze than simulated retrievals) is unclear. Thus, as currently written, this paper lacks important information needed by potential users for quantitative applications.

For readers of AMT, the manuscript should specifically address the following questions.

- What do the operational TROPOMI total column AKs look like, and how do they vary (for example, with respect to solar and satellite zenith angle, CO concentration, etc.)?

- Do the operational total column AKs depart from unity (i.e., the 'ideal case') far enough to be significant to users? In other words, is the smoothing error for operational TROPOMI CO column retrievals significant or not?

- For the operational TROPOMI retrievals, what are typical statistics for 'smoothing error' and 'retrieval noise' (as determined by the retrieval algorithm)? Do these quantities vary, or can they be fairly represented as fixed values?

Minor Revisions and Technical Corrections

Somewhere, the manuscript should clearly define the relationship between 'total column density of CO', which would have units of molecules per area, and the 'dry air mixing ratio' with units of ppb. Which of these is the 'actual' TROPOMI CO product?

Exactly where and how is the conversion between these two quantities made?

p. 1, line 8 - The adjective 'game-changing' should probably be replaced with a more formal term like 'groundbreaking'.

p. 2, line 8 - Please include a number for typical background CO concentrations after 'low background concentration'.

p. 2, line 21 - The SICOR algorithm exploits a CO 'reference profile' which is scaled to produce the retrieved CO total column. Is this reference profile a fixed profile, or does it vary geographically and/or seasonally? What method was used to obtain the reference profile?

p. 3, line 10 - The text is not clear regarding the criteria for classifying 'clear' and 'cloudy' cases, and which cases are discarded entirely. For example, are all cases where cloud optical thickness > 0.5 OR cloud height < 5000 considered cloudy, or is it really where cloud optical thickness > 0.5 AND cloud height < 5000? Please clarify.

p. 3, line 16 - The striping issue should be discussed in more detail, with a specific example (including a figure) describing the problem and demonstrating the improvement associated with the method. Does the '5 %' value refer to the bias in CO or to the number of retrievals affected by the issue?

p. 3, Section 3.1 - This section does not include any discussion or consideration of the effects of the different total column averaging kernels for TROPOMI and TCCON. As a result, the influence of the retrieval vertical sensitivity and smoothing error can not be distinguished from other conceivable types of retrieval bias. Therefore, in the title of Section 3.1 (and elsewhere in the manuscript), I would use the term 'Intercomparison' rather than 'Validation'. In-situ profiles are typically needed for proper validation work since retrieval vertical sensitivity and smoothing error can be explicitly represented.

p. 3, line 29 - The handling of discrepancies between the TCCON station altitude and the assumed altitude (or surface pressure) in the TROPOMI retrievals requires more

explanation, preferably with an example. Is it possible to estimate an uncertainty (or an upper limit) for TROPOMI/TCCON intercomparisons due to this effect? How does this uncertainty vary among the different TCCON sites?

p. 4, line 17 - The text implies, but does not clearly state, that neither temporal nor spatial averaging is necessary to '... to distinguish the CO enhancement of 20 ppb of the total column from the surrounding background concentrations ...' Please clarify. An analysis of TROPOMI retrieval noise (random error) would be an essential part of this discussion.

p. 4, line 18 - The text '20 ppb of the total column' should be reworded since ppb is not the unit for total column.

p. 6, line 25 - 'retrieval' is misspelled

Figures 1 and 2 include many data points for days where TROPOMI data were available but TCCON were not, or vice-versa. Such data were not used to calculate TROPOMI/TCCON bias statistics. I would suggest revising the figures somehow to identify which points were actually used to calculate the bias statistics.

---

## Referee Comment (RC2) · Anonymous Referee #2 · 24 Jun 2018

*Borsdorff et al. Mapping carbon monoxide pollution from space down to city scale with global coverage*

This short paper introduces the early validation of TROPOMI/Sentinel 5 Precursor XCO product with ground-based spectrometers using data during two months. The validation results indicate good accuracy and possibility for monitoring atmospheric CO globally on daily bases with relatively small spatial resolution, allowing thus further research on local sources of CO and the transport of the pollution in a novel way.

Overall the manuscript is well written and clear and the results are scientifically important. The manuscript is based on early TROPOMI observations and it is obvious that the aim is to report the validation results also in a timely manner. However, I have few general comments to encourage the authors to clarify what was actually done and to expand the discussion of the results. In addition, I have few minor comments.

**Major comments:**
1. In the abstract the game-changing nature of TROPOMI is emphasized. I would like to see further discussion on this topic perhaps in the Introduction chapter and later in the manuscript to more specifically address this point. E.g. it would be good to include some reference what the heritage instruments measured and why TROPOMI is a game-changer. Need for averaging data is briefly mentioned in Sec 3.2, but I would welcome a bit more discussion on this.
2. In the text both terms XCO and CO are used. Just by looking at the notation one might get the impression that XCO denotes daily values, which is perhaps not meant (Section 3.1). Please, clarify what is the TROPOMI data product and whether the validation was based on XCO or CO products.
3. In the validation both cloudy (low clouds) and cloud-free conditions are compared. Please, add discussion how valid FTS cloudy observations are, or clarify if only cloud free FTS observations are considered. Are TCCON and FTS in cloudy cases measuring the same air mass? Cloud optical thickness is used to select clear sky observations – where is this information coming from?
4. Please add a paragraph on FTS measurements since they are used as reference data here, their accuracy etc. The geographical distribution of the validation sites is limited to 50S-50N. Please, this is also good to be included in the text.
5. Soft calibration is done during the validation – please, discuss if this is also recommended when operational data is available.
6. Related to figures 4 and 6: Is here both cloudy data and cloud free data? Is there difference in the interpretation of the pixels depending on whether they are cloud free or cloudy? Please, add some discussion on this.
7. Conclusions: I would appreciate discussion on what exactly has been validated and to elaborate more what type of validation is needed in the future (in terms of spatial and temporal coverage, atmospheric and observational conditions).

**Minor comments:**

- Fig 1 & Fig 2. Please, indicate if this refers to cloud free data / cloudy data or both. If both, would it be possible to indicate this somehow.
- P 1, L 6: Sentence starting   Due to ….  – you could re-formulate this to make the message more clear.
- P 1, L 13:  Please, clarify station-to-station bias?
- P 2, L  13.  led -> lead ?
- Figure 5, lower panel, indications on how many observations correspond to varying radius values would be very nice to have.
- Fig 7 I think this is a zoom of figure 2. Is this needed?

---

## Author Comment (AC1) · 12 Jul 2018

We would like to thank reviewer 1 and 2 for the constructive comments that aided us to improve our manuscript. In this post we provide our replies to the reviewer's comments. We provide a revised version of the manuscript, in which all changes are highlighted. Revised and added text is provided in blue. In our replies to the comment we provide line numbers, page numbers and figure numbers of the old version of the manuscript.

Please also note the supplement to this comment:

[Figure]

https://www.atmos-meas-tech-discuss.net/amt-2018-132/amt-2018-132-AC1-supplement.pdf

[Figure]

**Supplement:**

**author comments on the manuscript amt-2018-132-RC1, reviewer 1**

We would like to thank reviewer 1 for the constructive comments that aided us to improve our manuscript. In this document we provide our replies to the reviewer's comments. The original comments made by the reviewer are numbered and typeset in italic and bold face font. Following every comment we give our reply. Here line numbers, page numbers and figure numbers refer to the original version of the manuscript, if not stated differently. Additionally, the revised version of the manuscript is added.

**General Comments**

1. *This brief manuscript reports early findings from CO retrievals by the TROPOMI instrument launched in late 2017. In that sense, the manuscript is similar to a GRL paper (Measuring Carbon Monoxide With TROPOMI: First Results and a Comparison With ECMWF-IFS Analysis Data) recently published by Borsdorff et al. The main novel aspect of this new manuscript is an intercomparison with ground-based CO measurements from nine TCCON stations. The manuscript also presents several short case studies demonstrating CO gradients associated with urbanization and long-range transport. Readers of Atmospheric Measurement Techniques may find this paper interesting, although it contains little technical information. Since the recent GRL paper by Borsdorff et al. already demonstrated that the TROPOMI instrument and SICOR retrieval algorithm yield physically reasonable CO retrievals, this manuscript should really go into the quantitative use of the TROPOMI CO product in some depth. Without such detailed technical information, users will not be able to fully exploit this valuable new dataset. Some of this information is contained in the 2014 AMT paper by Borsdorff et al. (Insights into Tikhonov regularization ...) and the 2016 AMT paper by Landgraf et al. (Carbon monoxide total column retrievals from TROPOMI ...), although the relevance of those theoretical papers to operational TROPOMI retrievals (which are inevitably more complicated to analyze than simulated retrievals) is unclear. Thus, as currently written, this paper lacks important information needed by potential users for quantitative applications.*

   Both the submitted manuscript and Borsdorff et al., GRL 2018 discuss the TROPOMI CO data product. Whereas the latter focuses on a first preliminary analysis of the product using a comparison with the ECMWF-IFS CO analysis data. The focus of this study is to quantify the uncertainty of the TROPOMI product by means of a comparison with ground-based reference measurements of the TCCON network. These measurements represent a well-accepted standard for the validation of satellite products, reported in several peer-reviewed papers. Most TCCON sites are located at remote regions and so the use of the NDACC site at Mexico City is another important and innovative element of the paper, demonstrating the validity of the product at regions of hot spot CO emissions. For users of the TROPOMI CO data, we consider such a study as essential, which quantifies the data quality. The technical details of the algorithm are well documented in the literature (Borsdorff et al., 2014 and Landgraf et al., 2016) has not changed since then, so we consider a proper referencing as sufficient. Based on the reviewers comments we see the clear need to discuss instrument specific aspects and the validation approach in more detail, which is described in our specific reply below.

2. *For readers of AMT, the manuscript should specifically address the following questions. What do the operational TROPOMI total column AKs look like, and how do they vary (for example, with respect to solar and satellite zenith angle, CO concentration, etc.)? Do the operational total column AKs depart from unity (i.e., the ideal case) far enough to be significant to users? In other words, is the smoothing error for operational TROPOMI CO column retrievals significant or not? For the operational TROPOMI retrievals, what are typical statistics for smoothing error and retrieval noise (as determined by the retrieval algorithm)? Do these quantities vary, or can they be fairly represented as fixed values?*

   **adjusted** We added the following discussion at p3,l13 and included a figure illustrating the vertical sensitivity of the retrieval and its dependence on the atmospheric state and on the observation geometry.

   "For the total column of CO, the vertical sensitivity of the retrieval is described by the total column averaging kernel (Borsdorff et al., 2014), which is illustrated in Fig. 1 for TROPOMI data of one particular day, 10th November 2017. It shows the dependence of the averaging kernel on the cloudiness of the scene, where the standard deviation indicates its variation due to different observation and atmospheric parameters, e.g. solar zenith angle, viewing zenith angle and ground reflectivity. For very strict cloud

clearing of the data (with $z < 5$ km and $\tau < 0.01$), the total column averaging kernel is close to 1 for all altitudes with little variation, meaning that the derived column can be interpreted as an estimate of vertically integrated amount of CO. Filtering the data less strict using the clear-sky filter from above ($z < 5$ km and $\tau < 0.5$) results in a slightly reduced sensitivity with a moderate standard deviation and Borsdorff et al. (2017) concluded that those measurements are usually clear-sky equivalent for remote regions without local pollution sources and the induced errors due to the choice of the reference profile to be scaled by the inversion to be on a percentage level (Borsdorff et al., 2014). The presence of clouds changes significantly the vertical sensitivity of the retrieval. Figure 1 shows the column averaging kernel when filtering for optical thick clouds at 5 km altitude. The sensitivity below the cloud is significantly reduced (values lower than 1) due to cloud shielding, and the retrieval estimates a CO total column mainly based on the measurement sensitivity to CO above the cloud (values higher than 1). This can lead to errors $> 30\%$ when the averaging kernel is not used for data interpretation (Borsdorff et al., 2014). However, the TROPOMI CO dataset provides total column averaging kernels for each retrieval and we recommend to use them when ever possible. ''

**Minor Revisions and Technical Corrections**

1. ***Somewhere, the manuscript should clearly define the relationship between total column density of CO, which would have units of molecules per area, and the dry air mixing ratio with units of ppb. Which of these is the actual TROPOMI CO product? Exactly where and how is the conversion between these two quantities made?***

   **adjusted** We added the following sentence on p3,l7:

   "Hence, the retrieval result is the total column density of CO [molec/cm$^2$]. To compare it with other measurements we also represent the data product as a dry column mixing ratio XCO [ppb] by dividing the CO total column density by the dry air column density derived from colocated ECMWF pressure fields. ''

2. ***p. 1, line 8 - The adjective game-changing should probably be replaced with a more formal term like groundbreaking.***

   **adjusted** We changed the term to "groundbreaking"

3. ***p. 2, line 8 - Please include a number for typical background CO concentrations after low background concentration.***

   **adjusted** We added a typical background value of CO at p2,l8:

   "With a typical background concentration of ca. 80ppb (in the Northern Hemisphere) . . . ''

4. ***p. 2, line 21 - The SICOR algorithm exploits a CO reference profile which is scaled to produce the retrieved CO total column. Is this reference profile a fixed profile, or does it vary geographically and/or seasonally? What method was used to obtain the reference profile?***

   **adjusted** We added the following sentence with a reference to the source of the reference profile:

   "The reference profile of CO that is scaled during the retrieval is taken from simulations of the global chemical transport model TM5 (Krol et al., 2005) and monthly averaged over 3 degree $\times$ 2 degree latitude/longitude grid boxes. ''

5. ***p. 3, line 10 - The text is not clear regarding the criteria for classifying clear and cloudy cases, and which cases are discarded entirely. For example, are all cases where cloud optical thickness ¿ 0.5 OR cloud height ¡ 5000 considered cloudy, or is it really where cloud optical thickness ¿ 0.5 AND cloud height ¡ 5000? Please clarify.***

   **adjusted** We changed the paragraph p3,l10 from:

   "Furthermore, we distinguished between retrievals under clear-sky ($\tau < 0.5$, $z < 5000$ m, over land) and cloudy condition ($\tau > 0.5$, $z < 5000$ m over land and ocean). ''

to

"Furthermore, we distinguished between retrievals under clear-sky ($\tau < 0.5$ and $z < 5$ km, over land) and cloudy condition ($\tau > 0.5$ and $z < 5$ km, over land and ocean). The remaining retrievals are not considered in this study. "

6. ***p. 3, line 16 - The striping issue should be discussed in more detail, with a specific example (including a figure) describing the problem and demonstrating the improvement associated with the method. Does the 5 % value refer to the bias in CO or to the number of retrievals affected by the issue?***

   **adjusted** We added a figure showing a typical striping pattern and changed the paragraph p3,l14 from:

   " The TROPOMI instrument is still in the early phase of the mission and the performance of the CO retrieval is expected to improve in the future. For example, single overpasses show stripes of erroneous CO values < 5% in the flight direction, probably due to calibration issues of TROPOMI. Considering high-frequency variations of CO measurements across flight direction per orbit, we infer the stripe pattern by median filtering of the detected features in flight direction. Subsequently, the stripe pattern is removed from the data. Boersma et al. (2011) suggested a similar approach to improve the quality of the $NO_2$ data product of the Ozone Monitoring Instrument (OMI, Levelt et al. (2006)). This aspect will be the subject of future investigations of the instrument calibration key data. "

   to

   " The TROPOMI instrument is still in the early phase of the mission and the performance of the CO retrieval is expected to improve in the future. For example, single overpasses show stripes of erroneous CO in flight direction, probably due to calibration issues of TROPOMI. Considering high-frequency variations of CO measurements across flight direction per orbit, we infer the stripe pattern by median filtering of the detected features in flight direction per orbit. Figure **??** provides an example, where the average the average of the stripe pattern in cross flight direction is -0.03 ppb with a standard deviation of 1.1 ppb. Some stripes can reach values higher than 5 ppb. Hence, the stripe pattern can be removed from the data a posteriori to the retrieval and its removal is indicated accordingly in the remainder of the paper. Boersma et al. (2011) suggested a similar approach to improve the quality of the $NO_2$ data product of the Ozone Monitoring Instrument (OMI, Levelt et al. (2006)). "

7. ***p. 3, Section 3.1 - This section does not include any discussion or consideration of the effects of the different total column averaging kernels for TROPOMI and TCCON. As a result, the influence of the retrieval vertical sensitivity and smoothing error can not be distinguished from other conceivable types of retrieval bias. Therefore, in the title of Section 3.1 (and elsewhere in the manuscript), I would use the term Intercomparison rather than Validation. In-situ profiles are typically needed for proper validation work since retrieval vertical sensitivity and smoothing error can be explicitly represented.***

   **adjusted** In general the reviewer is right. The TCCON columns are based on remote sensing and are affected by a smoothing error. However, Wunch et al. (2015) showed that the total error including the smoothing error of XCO is below 4%. The smoothing error of TROPOMI clear sky measurements is very similar and so within this uncertainty margin both data sets can be seen as an estimate of the true CO concentration for our study. We add the following sentence p3,l24:

   " Wunch et al. (2015) reported that the total error of the XCO columns measured by TCCON is below 4%. Hence, within this error margin we can assume the TCCON measurements as an estimate of the truth. "

8. ***p. 3, line 29 - The handling of discrepancies between the TCCON station altitude and the assumed altitude (or surface pressure) in the TROPOMI retrievals requires more explanation, preferably with an example. Is it possible to estimate an uncertainty (or an upper limit) for TROPOMI/TCCON intercomparisons due to this effect? How does this uncertainty vary among the different TCCON sites?***

**adjusted** We quantified the effect of the altitude correction and added the following paragraph at p3, l31:"

"The retrieved CO column of TROPOMI is adapted to the altitude of the station by either cutting off the scaled mixing ratios profile at the station altitude or extending it assuming a constant elongation of the mixing ratio to lower altitude. For mountain stations like Zugspitze and Izana, this reduces the TROPOMI CO column on average by 10 and 4 ppb, respectively, improving the agreement between theground-based and satellite measurements accordingly."

9. *p. 4, line 17 - The text implies, but does not clearly state, that neither temporal nor spatial averaging is necessary to ... to distinguish the CO enhancement of 20 ppb of the total column from the surrounding background concentrations ... Please clarify. An analysis of TROPOMI retrieval noise (random error) would be an essential part of this discussion.*

**adjusted** We calculated the retrieval noise error for the point sources shown in this study and add the following paragraph to the manuscript p4, l30:

" Neither temporal nor spatial averaging is necessary to distinguish the CO enhancements of the total column above the shown point sources. The average noise error of the retrievals from the individual ground pixels shown in Fig. 6 is $< 2.3$ ppb."

10. *p. 4, line 18 - The text 20 ppb of the total column should be reworded since ppb is not the unit for total column.*

**adjusted** We changed the sentence from :

" ... distinguish the CO enhancement of $\leq 20$ ppb of the total column from the surrounding background concentrations in the order of ..."

to
" ... distinguish typical the CO enhancements of $\leq 20$ ppb of the total column dry air mixing ratio from the surrounding background concentrations in the order of ...
"

11. *p. 6, line 25 - retrieval is misspelled Figures 1 and 2 include many data points for days where TROPOMI data were available but TCCON were not, or vice-versa. Such data were not used to calculate TROPOMI/TCCON bias statistics. I would suggest revising the figures somehow to identify which points were actually used to calculate the bias statistics.*

**adjusted** we recalculated the Fig. 1 and 2. Data without time coincidence for TCCON or TROPOMI is plotted in grey. Furthermore we added the following sentence to the figure caption of Fig.1:

[revised manuscript text omitted]

---

## Author Comment (AC2) · 12 Jul 2018

We would like to thank reviewer 1 and 2 for the constructive comments that aided us to improve our manuscript. In this post we provide our replies to the reviewer's comments. We provide a revised version of the manuscript, in which all changes are highlighted. Revised and added text is provided in blue. In our replies to the comment we provide line numbers, page numbers and figure numbers of the old version of the manuscript.

Please also note the supplement to this comment:

[Figure]

https://www.atmos-meas-tech-discuss.net/amt-2018-132/amt-2018-132-AC2-supplement.pdf

[Figure]

**Supplement:**

**author comments on the manuscript amt-2018-132-RC1, reviewer 2**

We would like to thank reviewer 2 for the constructive comments that aided us to improve our manuscript. In this document we provide our replies to the reviewer's comments. The original comments made by the reviewer are numbered and typeset in italic and bold face font. Following every comment we give our reply. Here line numbers, page numbers and figure numbers refer to the original version of the manuscript, if not stated differently. Additionally, the revised version of the manuscript is added.

**Major Comments**

1. *In the abstract the game-changing nature of TROPOMI is emphasized. I would like to see further discussion on this topic perhaps in the Introduction chapter and later in the manuscript to more specifically address this point. E.g. it would be good to include some reference what the heritage instruments measured and why TROPOMI is a game-changer. Need for averaging data is briefly mentioned in Sec 3.2, but I would welcome a bit more discussion on this.*

    **adjusted** We add the following text

    p2,l5:

    "…and a high radiometric accuracy to infer the CO total column over dark vegetation surfaces with an precision < 10 % (Veefkind et al., 2012)"
    and p2, l11
    "The S5P mission builds upon the heritage of SCIAMACHY (Scanning Imaging Absorption Spectrometer for Atmospheric Chartography; Bovensmann et al. (1999)), which provided atmospheric CO total column concentrations from the same spectral range (Borsdorff et al., 2017a, Buchwitz et al., 2007, Frankenberg et al., 2005, Gimeno Garcia et al., 2011, Gloudemans et al., 2009)). Measurements of SCIAMACHY in the SWIR have a spatial resolution of about 30 km × 120 km (along-track × across-track) for an integration time of 0.5 s with a global coverage cycle of 3 days. Most importantly, the SCIAMACHY noise error of single CO retrievals can exceed 100 % for a dark scenes. Hence, spatial and temporal averaging of single CO measurements is required (de Laat et al., 2007, Gloudemans et al., 2006), which limits the data interpretation of SCIAMACHY CO data."

2. *In the text both terms XCO and CO are used. Just by looking at the notation one might get the impression that XCO denotes daily values, which is perhaps not meant (Section 3.1). Please, clarify what is the TROPOMI data product and whether the validation was based on XCO or CO products.*

    **adjusted** We added the following sentence on p3,l7:

    "Hence, the retrieval result is the total column density of CO [molec/cm$^2$]. To compare it with other measurements we also represent the data product as a dry column mixing ratio (XCO, [ppb]) by dividing the CO total column density by the dry air column density. "

3. *In the validation both cloudy (low clouds) and cloud-free conditions are compared. Please, add discussion how valid FTS cloudy observations are, or clarify if only cloud free FTS observations are considered. Are TCCON and FTS in cloudy cases measuring the same air mass? Cloud optical thickness is used to select clear sky observations where is this information coming from?*

    **adjusted** We added the following sentence on p3,l25:

    " Cloud contaminated measurements are rejected and so TCCON measurements refer to clear-sky observations only. Here, TCCON CO columns are provided as column averaged dry air mole fractions XCO (Wunch et al., 2010). "

4. *Please add a paragraph on FTS measurements since they are used as reference data here, their accuracy etc. The geographical distribution of the validation sites is limited to 50S-50N. Please, this is also good to be included in the text.*

**adjusted** We added the following paragraph on p3,l25:

"TCCON is a network of ground-based Fourier Transform Spectrometers to measure total column concentrations of atmospheric trace gases including CO with high accuracy and precision e.g. for satellite validation. The trace gas columns are retrieved from spectrally highly resolved near-infrared radiance measurements recorded in direct-sun geometry (Wunch et al., 2015)"

5. ***Soft calibration is done during the validation  please, discuss if this is also recommended when operational data is available.***

    **adjusted** The stripe correction mentioned by the reviewer was not applied during the validation. This is an important point that was not mentioned in the manuscript before. We added the following sentence on p3,l14:

    "It is important to note that we only applied the de-striping approach on the Figs. 6 and 8 but not on the TROPOMI CO data used in the later validation. The CO data striping will be the topic of upcoming studies with the first preference to improve the instrument calibration key data."

6. ***Related to figures 4 and 6: Is here both cloudy data and cloud free data? Is there difference in the interpretation of the pixels depending on whether they are cloud free or cloudy? Please, add some discussion on this.***

    **adjusted** In Figure 4 and 6, we do not distinguish between clear-sky and cloudy data to obtain a good data coverage. For clarification, we added the following sentence at p4, l29

    "Figure 6 shows predominately clear-sky observations but also includes retrievals from cloud contaminated scenes, which in case of optical thick and high clouds reduces the sensitivity to boundary layer CO pollution at emission hot spots (Borsdorff et al., 2017b)."

7. ***Conclusions: I would appreciate discussion on what exactly has been validated and to elaborate more what type of validation is needed in the future (in terms of spatial and temporal coverage, atmospheric and observational conditions).***

    **adjusted** We already state in the conclusion (p6,l10-14) that the TROPOMI CO dataset (clear-sky, cloudy) was validated with TCCON and NDACC measurements. Furthermore, we reported the biases with the validation measurements. We add the following sentence at p6,l13:

    " For this study, only a limited amount of TCCON data was available with confined spatial and temporal coverage. The Sentinel 5 Precursor as an operational mission requires a continuous monitoring of the CO data quality, which will be performed as part of the operational validation activities. In this context, future work will consider the validation of the TROPOMI CO data for longer time scales including additional TCCON and NDACC stations to improve the significance of the product validation."

**Minor comments**

1. ***Fig 1 & Fig 2. Please, indicate if this refers to cloud free data / cloudy data or both. If both, would it be possible to indicate this somehow.***

    **adjusted** The daily means in the plots include both clear-sky and cloudy satellite data. We indicated this in the caption of Fig.1 by adding the sentence:

    " Clear-sky and cloudy satellite data are used to calculate the daily means. "

2. ***P 1, L 6: Sentence starting Due to ....  you could re-formulate this to make the message more clear.***

    **adjusted** We change the sentence p1,l6 from:

    " Due to its moderate atmospheric residence time, its atmospheric abundance provides information on both localized pollution hot spots and the pollutant transport on regional to global scales. "
    to

" The moderate atmospheric resistance time and the low background concentration leads to localized pollution hot spots of CO and allows to track the atmospheric transport of pollution on regional to global scales.
"

3. **P 1, L 13: Please, clarify station-to-station bias?**

   **adjusted** We changed the sentence at p1,l13 from:

   " We found a good agreement between both data sets with a mean bias of 6 ppb for both clear-sky and cloudy TROPOMI CO retrievals. Together with the corresponding standard deviation of the station-to-station bias of 3.9 ppb for clear-sky and 2.4 ppb for cloudy-sky, it indicates that the CO data product is already well within the mission requirement."
   to
   " We found a good agreement between both data sets with a mean bias of 6 ppb (average of individual station biases) for both clear-sky and cloudy TROPOMI CO retrievals. Together with the corresponding standard deviation of the individual station biases of 3.8 ppb for clear-sky and 4.0 ppb for cloudy-sky, it indicates that the CO data product is already well within the mission requirement.
   "

4. **P2,L 13. led-¿lead?**

   **corrected**

5. **Figure 5, lower panel, indications on how many observations correspond to varying radius values would be very nice to have.**

   **adjusted** We add the following two sentences in the caption of Fig. 5:

   " For the smallest radius (15 km) we found 20 cloudy and 160 clear-sky collocations. However, for the widest radius (100 km) 92 cloudy and 4425 clear-sky collocations are found. "

6. **Fig 7 I think this is a zoom of figure 2. Is this needed?**

   **adjusted** The reviewer is right we removed Fig. 7 from the manuscript.

**References**

[revised manuscript text omitted]

---

## Author Response (AR2)

**author comments on the manuscript amt-2018-132-RC1, reviewer 1**

We would like to thank reviewer 1 for the constructive comments that aided us to improve our manuscript. In this document we provide our replies to the reviewer's comments. The original comments made by the reviewer are numbered and typeset in italic and bold face font. Following every comment we give our reply. Here line numbers, page numbers and figure numbers refer to the original version of the manuscript, if not stated differently. Additionally, the revised version of the manuscript is added.

1. *Following the sentence "However, the TROPOMI CO dataset provides total column averaging kernels for each retrieval and we recommend to use them when ever possible" in Section 2, please provide clear instructions to potential users (including the relevant equation) regarding the actual method for applying the total column averaging kernel to CO in-situ datasets or model simulations.*

   **adjusted** We changed the paragraph in section 2 at p4, l2 form:

   " This can lead to errors $> 30\%$ when the averaging kernel is not used for data interpretation (Borsdorff et al., 2014). However, the TROPOMI CO dataset provides total column averaging kernels for each retrieval and we recommend to use them when ever possible. "

   to

   " Consequently, the direct comparison of reference measurements with the retrieved CO columns from cloud contaminated TROPOMI measurements can lead to errors $> 30\%$ (Borsdorff et al., 2014). This so called smoothing error is due to imperfect knowledge of the vertical profile of CO. However, the TROPOMI CO dataset provides total column averaging kernels $a_col$ for each retrieval. To compare a vertical profile $\rho$ e.g. from airborne in-situ measurements or model simulations with the TROPOMI CO product a total column concentration $c = a_{col}\dot\rho$ can be calculated from $\rho$. This can be directly compared with the retrieval result since it is in the same way affected by the reduced sensitivity as the retrieval (Rodgers and Connor, 2003). When the reference measurement is not a vertical profile the application of the total column averaging kernel becomes more difficult. In that case, the TROPOMI CO dataset can be filtered for retrievals under clear-sky conditions to avoid misinterpretations. Alternatively, an approach as presented by Cogan et al. (2012) can be followed who quantified expected differences in GOSAT/TCCON $CO_2$ retrievals due to averaging kernel differences using the GEOS-Chem model to simulate a realistic range of $CO_2$ profiles. "

2. *The similarity of clear- and cloudy-scene validation statistics at the end of Section 3.1 might imply to readers that these two subsets are equivalent with respect to data quality and can be used interchangeably. However, the similarity in validation statistics could simply result from the selection of scenes where the under-cloud CO profile was generally consistent with the above-cloud CO profile (in terms of profile shape). This would not be the case in strong CO source regions, for example. Please re-emphasize to the readers at the end of Section 3.1 that clear-sky retrievals are always preferable (compared to cloudy-sky retrievals) because of the more consistent sensitivity to CO over the entire profile.*

   **adjusted** We added the following paragraph in section 3.1 at p5,l11:

   " Most of the TCCON stations are only affected by remote pollution sources, this explains the good agreement between the validation of the clear-sky and cloudy-sky TROPOMI retrievals. This may differ in the presence of local pollution sources where the shape of the under-cloud CO profile can strongly deviate from the one of the reference profile used for the profile scaling of the TROPOMI CO dataset. In such cases, when the TROPOMI CO data set is directly compared with reference measurements without applying the averaging kernel clear-sky are always preferable compared to cloudy-sky observations. "

3. *Since the validation results reported in this paper largely hinge on the absolute accuracy of the TCCON retrievals, please provide stronger quantitative evidence for the effects of smoothing error on TCCON CO retrievals, preferably from the peer-reviewed literature. If the effects of smoothing error as a source of retrieval error for the TCCON retrievals have not been explicitly studied, that should be stated in the manuscript.*

**adjusted** We changed the following paragraph at p4,ll:

[revised manuscript text omitted]